# Bovine leukemia virus encoded blv-miR-b4-3p microRNA is associated with reduced expression of anti-oncogenic gene *in vivo*

**Marcos I. Petersen**[ID]◉, **Hugo A. Carignano**◉, **Claudia Mongini, Diego D. Gonzalez, Juan P. Jaworski**[ID]*

Instituto de Virología e Innovaciones Tecnológicas (IVIT), Instituto Nacional de Tecnología Agropecuaria (INTA), Consejo Nacional de Investigaciones Científicas y Tecnológicas (CONICET), Buenos Aires, Argentina

◉ These authors contributed equally to this work.
* jaworski.juan@inta.gob.ar

**Data Availability Statement:** All relevant data are within the paper and its Supporting Information files.

## Abstract

Bovine leukemia virus (BLV) is a retrovirus that causes malignant B-cell lymphoma in up to ten-percent of infected cattle. To date, the mechanisms of BLV linked to malignant transformation remain elusive. Although BLV-encoded miRNAs have been associated with the regulation of different genes involved in oncogenic pathways, this association has not been evaluated in cattle naturally infected with BLV. The objective of this study was to determine the relative expression of BLV-encoded miRNA blv-miR-b4-3p, the host analogous miRNA bo-miR-29a and a couple of potential target mRNAs (HBP-1 and PXDN, with anti-tumorigenic function in B-cells), in cattle naturally infected with BLV compared to uninfected animals (control group). We observed that PXDN was significantly downregulated in BLV-infected cattle (P = 0.03). Considering the similar expression of endogenous bo-miR-29a in both animal groups, the downregulation of PXDN in BLV-naturally infected cattle could be linked to blv-miR-b4-3p expression in these animals. Knowing that PXDN is involved in anti-tumoral pathways in B-cells, the results presented here suggest that blv-miR-b4-3p might be involved in BLV tumorigenesis during natural infection with BLV in cattle.

## Introduction

Bovine leukaemia virus (BLV) belongs to the Deltaretrovirus genus, Orthoretrovirinae subfamily and Retroviridae family. BLV is related to the human T-cell lymphotropic virus (HTLV-1 and -2). According to the Baltimore classification system BLV is a +ssRNA virus (Group IV [1]. BLV infects mainly B-lymphocytes (B-cells) CD5+ and once inside the cell, the viral genome is rapidly retro-transcribed and the resulting genomic DNA (gDNA) is inserted randomly within the host cell genome. During the acute stage of BLV infection, virus spread occurs via the production of virions able to infect new target cells (replicative stage) [2]. The synthesis of viral transcripts and proteins is a hallmark of this stage. The expression of viral antigens induces a strong immune response, which effectively limits the virus´ productive

**Funding:** JPJ: PICT-2017-0262. Fondo para la Investigación Científica y Tecnológica (FONCyT). https://www.argentina.gob.ar/ciencia/agencia/fondo-para-la-investigacion-cientifica-y-tecnologica-foncyt. The funders had no role in study design, data collection and analysis, decision to publish, or preparation of the manuscript.

**Competing interests:** The authors have declared that no competing interests exist.

cycle. Since anti-viral immune response persists throughout the animal's life, those viral clones with low-expression are selected and expand by mitosis of host cells carrying BLV provirus by a process known as clonal expansion (mitotic infection stage/ chronic stage) [2].

BLV causes a persistent infection in cattle and in most cases this infection is asymptomatic. In one third of infected animals the infection progresses to a state of persistent lymphocytosis (PL) and 1 to 10% of infected cattle develop leukemia and lymphosarcoma (LS) [3]. The tumorigenic mechanisms induced by BLV remain largely unknown. Although the production of viral proteins is abolished during the chronic stage, viral transcription is not completely shut down. Different groups have shown that BLV genome encodes for different micro-RNAs (miRNAs) which have the potential to regulate different host metabolic routes and modulate pathogenesis [4–9]. miRNAs are small (19–24 bases) non-coding RNAs with posttranscriptional regulatory capacity [10]. miRNAs bind through its seed sequence (up to 9 bases located 5′) to messenger RNAs (mRNAs) promoting its degradation and/or impeding translation. miRNAs are produced by eukaryotic (host) cells and more recently, viral-encoded miRNAs have also been reported [10].

One of BLV-encoded miRNAs (blv-miR-b4-3p) shares 9 nucleotides of its seed region with the host miR-29a [4,5]. miR-29a targets HMG-box transcription factor 1 (HBP-1) and peroxidasin homolog (PXDN) mRNAs reducing their expression [4,5]. Both HBP-1 and PXDN gene products have anti-tumoral activity in B-cells. Endogenous miR-29a overexpression is linked to B-cell malignancies in humans [11]. Additionally, endogenous and exogenous overexpression of miR-29a induces B-cell tumors in mice [12,13]. Considering the oncogenic role of miR-29a, it is possible that blv-miR-b4-3p could be involved in BLV tumorigenesis mechanisms. In this regard, a couple of studies showed *in vitro* and *ex vivo* that blv-miR-b4-3p downregulated the expression of HBP-1 and PXDN [4,5]. One of these studies also reported high *in vivo* expression of blv-miR-b4-3p in an ovine model for BLV infection [5].

According to these observations, we hypothesize that the expression of BLV-encoded blv-miR-B4-3p in cattle naturally infected with BLV, reduces the expression of HBP-1 and PXDN, while host-miRNA bo-mir-29a expression is kept at basal levels. To test our hypothesis, we used in-house RT-qPCRs for the quantification of viral-encoded miRNA blv-miR-b4-3p, host-encoded miRNA bo-miR-29a and target mRNAs (HBP-1 and PXDN) in cattle naturally infected with BLV.

## Materials and methods

### Animals, sampling and PBMCs isolation

Six Holstein cows from the National Institute of Agricultural Technology (INTA, AR) were characterized regarding their BLV status. Individuals aged approximately 4 years old belonged to the main dairy productive area in Argentina (31°16′S, 61°29′W), where the mean prevalence of BLV is higher than 80%. All animals were phenotypically balanced (i.e., sex, age, physiological status, weight, etc.). BLV infection status in these animals was assessed by the presence of BLV-genomic DNA (pol) and BLV-specific Abs by qPCR and ELISA, respectively.

In order to avoid RNA degradation and changes *ex vivo*, sample collection, peripheral blood mononuclear cells (PBMCs) isolation and conservation in RNAlater™ solution (Ambion, Austin, US) were performed within the same working day (T0), as was previously described [14]. In short, blood was collected from the jugular vein using as anticoagulant EDTA solution (225 μM). PBMCs were isolated with Ficoll-Paque PLUS (GE Healthcare, Uppsala, Sweden) density-gradient centrifugation following the manufacturer's protocol. After centrifugation plasma fractions were stored at -20° C until they were used for assessment of anti-BLV antibodies by ELISA.

Animal handling and sampling procedures followed recommendations of the Institutional Animal Care and Use Committee of the National Institute of Agricultural Technology.

## Anti-BLV ELISA

Indirect ELISA against the whole lysed BLV viral particle was used [15]. Briefly, a weak positive control serum was used as reference to calculate a normalized sample to positive ratio called percentage of reactivity (PR). The difference between the OD value obtained for the weak positive control and a negative BLV control sample was set up as 100% of reactivity. All tested samples were referred to it. Those samples with PR above the cut-off level (>25%) were considered positive.

## Nucleic acid purification

Genomic DNA (gDNA) for determination of BLV-status by qPCR was extracted from PBMCs using High Pure PCR Template Preparation Kit (Roche, Penzberg, Germany) according to the manufacturer's instructions. Concentration was measured at 260 nm wavelength using a nanophotometer (Nanodrop, ThermoFisher Scientific, Waltham, United States); moreover, DNA purity was assessed by ratio of absorbance at 260/230 nm and 260/280 nm.

Total RNA, including small fragments for quantification of messenger RNA (mRNA) and micro-RNA (miRNA), was extracted with the miRNeasy Mini Kit (Qiagen, Düsseldorf, Germany). An on-column DNase (Qiagen) digestion step was added following manufacturer´s recommendation to avoid downstream carryover of gDNA. RNA concentration and purity were measured using a nanophotometer (Nanodrop, ThermoFisher Scientific, Waltham, US) and quality was assessed using a Bioanalyzer (Agilent, Santa Clara, US).

## BLV-pol qPCR

BLV-pol DNA was detected using a previously validated in-house qPCR technique [16,17]. Briefly, each qPCR reaction (final volume Vf = 25 μl) contained 1x Fast Start Universal SYBR Green Master Mix (Roche, Penzberg, Germany), Forward and Reverse primers (800 nM) (BLVpol_5f: 5′-CCTCAATTCCCTTTAAACTA-3′; BLVpol_3r: 5′-GTACCGGGAAGACTG GATTA-3′, respectively) and gDNA template (200–1000 ng). The amplification and detection reaction were performed on a Step One Plus equipment (Applied Biosystem, Foster City, US). The specificity of each BLV positive reaction was confirmed by melting temperature dissociation curve (Tm) analysis. Previously described standard DNA, positive and negative controls were used. Values > 1500 copies per μg of total gDNA (copies/μg) were considered as high PVL [14].

## Determination of mRNA expression by RT-qPCR

A quantity of 1000 ng of RNA was used for cDNA synthesis using MultiScribe™ High capacity cDNA Retro-Transcription (RT) kit (Applied Biosystem, Foster City, US) according to manufacturer's instructions. Briefly, the RNA was added to a master mix (Vf = 20 μl), containing 2 μl of 10X RT buffer, 2 μl of 10X RT Random Primers, 4 mM dNTPs, 50 U Multiscribe™ reverse transcriptase, 10 U RNasin® Ribonuclease Inhibitors (Promega, Madison, US) and ultra-pure water (Invitrogen, Waltham, US). Cycling conditions were as follows: i) 25˚C for 10 min, ii) 37˚C for 120 min and iii) 85˚C for 5 min. A 1:5 dilution of cDNA in ultra-pure water (Invitrogen) was used in each of the qPCR described below.

BLV-pol expression was assessed with the BLV-pol qPCR described above. Each sample was run in triplicate, including 1 no RT replicate as a gDNA contamination control. A positive

**Table 1. Primers designed for candidate target mRNA and reference genes amplification (qPCR).**

| Gen[1] | | Sequence 5´-3´ |
|---|---|---|
| HBP1 | Fw | GCGACGGGTTTGTCGGA |
| | Rv | TATCCAGGAGAGGACGGCAA |
| PXDN | Fw | CCCACCTTGATGTGTCTGCT |
| | Rv | GTGATGTTGTCGGCGTTGTC |
| RPLP0 | Fw | CAACCCTGAAGTGCTTGACAT |
| | Rv | AGGCAGATGGATCAGCCA |
| B2M | Fw | AGCAAGGATCAGTACAGCTGCCG |
| | Rv | ATGTTCAAATCTCGATGGTGCTGCT |

control and a negative control, as well as two no-template control (RT-NTC and qPCR-NTC), were included in each run.

Primers for both mRNA (HBP-1 and PXDN) targeted by miR-29a and reference/housekeeping genes (RPLP0 and B2M) were designed using the Primer-BLAST tool [18]. Self-dimer and hetero-dimer tendencies, GC content, melting temperature (Tm) and potential secondary structure for each primer pair were evaluated with the OligoAnalyzer 3.1v software [19]. The qPCR reaction was performed with the Fast Start Universal SYBR Green Master Mix (Roche, Penzberg, Germany). The final reaction volume was set at 10 μl, adding 3 μl of cDNA dilution and 300 nM HBP-1 or 300 nM RPLP0 or 300nM B2M or 150 nM PXDN primers (Table 1). The qPCR reactions were conducted in StepOne Plus device (Applied Biosystem) following a standard running condition: i) activation step at 95°C, 10 min; ii) 40 cycles of 95°C for 15 s and 60°C for 1 min. The specificity of each reaction was confirmed by melting temperature (Tm) dissociation curve analysis and agarose gel electrophoresis. Technical duplicates were assayed for all RT-qPCR reactions in two separate runs. For each evaluated gene, no retro-transcribed RNA template and no template controls were included in each reaction.

The reaction efficiencies were calculated using the automated approach of LinRegPCR software [20], which performs baseline correction and linear regression analysis on individual sample amplification curve. Relative expression values were normalized against previously described bovine reference genes: Ribosomal protein large P0 (RPLP0) and β-2-Microglobulin (B2M) [21].

## Quantification of miRNAs by stem-loop RT-qPCR

A quantity of 1000 ng of RNA was utilized for cDNA synthesis using MultiScribe™ High capacity cDNA Retro-Transcription (RT) kit (Applied Biosystem, Foster City, US). The manufacturer's protocol was modified using custom designed stem-loops primers specific to each miRNA (Table 2). Each miRNA was retro-transcribed in a separate reaction. Briefly, RT reactions contained 1000 ng of purified RNA, 50 nM of specific stem-loop RT primer, 1x RT buffer, 4mM dNTPs, 25 U Multiscribe™ reverse transcriptase, 5 U RNasin® Ribonuclease Inhibitors (Promega, Madison, US) and ultra-pure water (Invitrogen, Waltham, US), in a total volume of 10 μl. Cycling conditions were as follows: i) 16°C for 30 min, ii) 42°C for 30 min and iii) 85°C for 5 min. No-template controls and RT-minus controls were included in each reaction. A 1:5 dilution of cDNA in ultra-pure water (Invitrogen) was used for qPCR.

Custom forward (Fw) primers were designed for each miRNA target and a unique reverse (Rv) primer was used in both reactions (Table 2). For qPCR, we used the Fast Start Universal SYBR Green Master Mix (Roche). For miR-29a qPCR, the final reaction volume was 12.5 μl, adding 1x buffer, 2.5 μl of cDNA dilution, 500 nM Fw and Rv primers, and 100 nM DMSO.

**Table 2. Primers designed for STEM-LOOP RT and miRNA amplification (qPCR).**

| Name | Sequence 5´-3´ |
|---|---|
| STEM-LOOP-miR-b4 | GTCTCCTCTGGTGCAGGGTCCGAGGTATTCGCACCAGAGGAGACAAAGGC |
| STEM-LOOP-miR-29a | GTCTCCTCTGGTGCAGGGTCCGAGGTATTCGCACCAGAGGAGACTAACCG |
| miR-b4-Fw | GGCGGTAGCACCACAGTCTCT |
| miR-29a-Fw | GGCGGTAGCACCATCTGA |
| SL-Rv | GTGCAGGGTCCGAGGTATT |

For blv-miR-b4-3p qPCR, the final reaction volume was 12.5 µl, adding 1x buffer, 2.5 µl of cDNA dilution, 5000 nM Fw primer and 1000 nM Rv primer. The qPCR reactions were conducted in StepOne Plus device (Applied Biosystem) following a standard running condition: i) activation step at 95˚C, 10 min; ii) 40 cycles of 95˚C for 15 s and 60˚C for 1 min. The reaction for blv-miR-b4-3p had an additional annealing step at 55˚C, for 10 s. The specificity of each reaction was confirmed by melting temperature (Tm) dissociation curve analysis. Technical duplicates were assayed for all RT-qPCR reactions in two separate runs. No retro-transcribed RNA template and no template controls were included in each qPCR reaction. In addition, non-specific retro-transcribed cDNA (e.g., cDNA retro-transcribed with mir-29a stem-loop primer was assayed in blv-miR-b4-3p qPCR and vice versa) was included for specificity control.

The reaction efficiencies were calculated using the automated approach of LinRegPCR software [20]. Relative expression values were normalized against bovine reference genes RPLP0 and B2M [21].

## RT-qPCR statistical analyses

All RT-qPCR assays were carried out following the recommendations proposed by the MIQE (Minimum Information for Publication of Quantitative Real-Time PCR Experiments) guidelines [22]. Differential target gene expression (mRNA and miRNA), was evaluated as previously described [14]. Briefly, analysis between positive and negative animals, was measured using the $N_0$ values (Eq 1) [23]:

$$N_0 = \frac{N_t}{E^{C_q}} \qquad \text{Eq 1}$$

Where $N_0$ = initial template concentration measured in fluorescence values; $N_t$ = user defined fluorescence threshold; E = reaction efficiency and $C_q$ = fractional number of cycles needed to reach the established threshold.

Normalization of target genes $N_0$ was done against the geometric mean of $N_0$ values from RPLP0 and B2M reference genes [24]. This normalization process was done for the cDNA of each animal. The normalized $N_0$ values were transformed to a based-10 logarithmic scale to approximate a normal distribution. Once normal distribution was confirmed (Shapiro-Wilk test), we performed the unpaired T-test to evaluate differential gene expression between BLV (+) and control (BLV-) groups, using GraphPad Prism (Dotmatics, MA, US) with a 95% confidence interval.

## Results

### Animal groups description

BLV prevalence in adult dairy cattle in Argentina is above 80%. In order to make a proper characterization regarding their BLV-infection status, we followed six adult Holstein cows that

**Table 3. Animal characterization regarding BLV infection.**

| ID | ELISA anti-BLV (%)[a] | | | | | BLV-pol qPCR (cop./μg gDNA)[b] | | BLV-pol RT-qPCR[c] |
|---|---|---|---|---|---|---|---|---|
| | T(-10) | T(-5) | T(-3) | T(0) | T(3) | T(-3) | T(0) | T(0) |
| 6462 | ND | ND | ND | ND | ND | ND | ND | ND |
| 6493 | ND | ND | ND | ND | ND | ND | ND | ND |
| 5830 | ND | ND | ND | ND | ND | ND | ND | ND |
| 5841 | 180 | 169 | 179 | 180 | NA | 33660 | 54000 | -3.71 |
| 6021 | 184 | 168 | 168 | 118 | NA | 17000 | 87000 | -2.72 |
| 6097 | 188 | 178 | 185 | 166 | NA | 80000 | 102000 | -2.93 |

[a] Reactivity percentage of ELISA.

[b] Proviral load (PVL) as copies per μg of genomic DNA (gDNA).

[c] BLV pol normalized expression ($Log_{10} N_0$).

ND: Not detected.

NA: Not assayed.

were in production for a period of 1 year. Natural infection with BLV in three of these animals was determined by the detection of BLV-specific antibodies by ELISA and the detection of BLV proviral (pol) DNA by BLV-pol qPCR tests. All three naturally-infected cattle [BLV (+)] presented a high proviral load (> 1500 copies/μg) measured at T(-3) and T0 (Table 3). Interestingly, we also detected the expression of high levels of BLV-pol mRNA in these three BLV (+) animals at T0 (Table 3 and Fig 1A).

The BLV (-) status in the remaining 3 animals was determined by the absence of BLV-specific antibodies and BLV genomic DNA. To confirm that BLV (-) animals did not seroconvert around the sampling time-point (T0), the seronegative status in all 3 BLV (-) animals was confirmed 3 months after the sampling day (T3). No BLV-sp antibodies and no BLV pol DNA were detected in these animals at any tested time-point (Table 3).

## PXDN downregulation by virus-encoded blv-miR-b4-3p in cattle naturally infected with BLV

We measured the expression of blv-miR-b4-3p and the host-encoded analogous bo-miR-29a, in three BLV naturally infected and three non-infected cattle (control group). Both host and viral miRNAs share 9 nucleotides within their seed region indicating common effector targets (mRNA). As expected, blv-miR-b4-3p was detected in BLV (+) animals only (Fig 1B). Oppositely, we observed no difference in miR-29a expression between BLV (+) and BLV (–) animals (Fig 1C; P = 0.2043). Interestingly, blv-miR-b4-3p levels detected in BLV (+) animals were more than 1 Log higher (> ten-folds) compared to endogenous bo-miR-29a.

In addition, we measured the expression and tested the potential downregulation of two mRNA (HBP-1 and PXDN) that have been previously proposed as potential targets of miR-29a. We observed that PXDN expression was significantly lower in BLV (+) cattle compared to BLV (-) cattle (Fig 1D; P = 0.0317). In contrast, we did not observe variation in HBP-1 expression levels between BLV (+) and BLV (-) cattle (1.E; P = 0.4237).

## Discussion

BLV is an oncogenic retrovirus affecting cattle worldwide. Since the discovery of BLV-encoded miRNAs, efforts to elucidate their role in viral pathogenesis have been conducted. Particularly, blv-mir-b4-3p mimics host miRNA mir-29a, a suspected oncomiR targeting PXDN and HBP-1. Interestingly, PXDN and HBP-1 have been involved with B-cell tumor suppression in

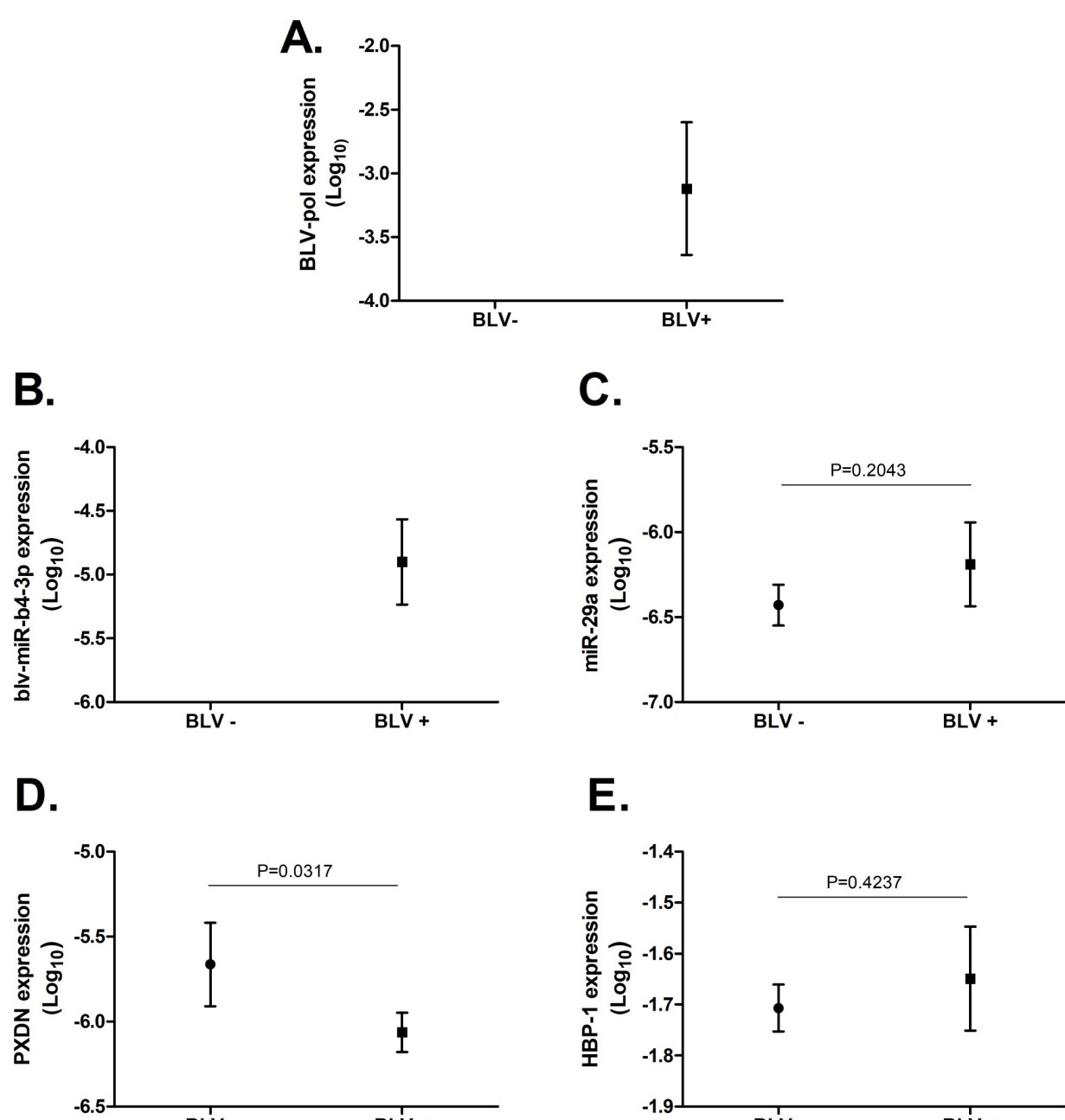

**Fig 1. Expression of viral and host miRNAs and mRNAs.** Normalized expression of **A.** BLV-pol, **B.** viral blv-miR-b4-3p, **C.** host miR-29a, **D.** PXDN mRNA and **E.** HBP-1 mRNA in BLV (+) and BLV (-) cattle. Relative quantification of each particular target was performed by RT-qPCR following MIQE guidelines (see Materials and Methods section). Normalized expression levels were compared between BLV-infected (n = 3) and BLV-uninfected (n = 3) cattle using unpaired T-test; p values are stated within the graph. Mean value (dot) and standard deviation (vertical bar) are indicated for each animal group.

different model organisms (i.e., mice) and humans [7,11–13]. Differential expression of such target genes *in vivo* during natural infection with BLV remains largely unexplored.

In the present study, we describe for the first time an association between the presence of BLV-encoded miRNA blv-miR-b4-3p and a reduction of PXDN mRNA levels in cattle naturally infected with BLV. Considering that levels of endogenous bo-miR-29a were similar in

BLV (+) and BLV (-) animals, we assume that at least in part, downregulation of PXDN in BLV naturally infected cattle is a consequence of blv-miR-b4-3p expression. Moreover, since PXDN has been linked to anti-tumoral pathways in B-cells, the downregulation of PXDN expression by blv-miR-b4-3p might be involved in BLV tumorigenesis in cattle during natural infection with this virus.

Our finding is in agreement with previous reports pointing-out the posttranscriptional downregulation of PXDN by blv-miR-b4-3p *in vitro* using reporting genes [4]. In the same work, the authors reported that blv-miR-b4-3p also targeted HBP-1, another anti-tumoral gene, *in vitro*. However, in our study we did not find any association between blv-miR-b4-3p expression in cattle naturally infected with BLV and HBP-1 mRNA expression levels, compared to the control group (BLV-). In agreement with our observation, Ochiai *et al.* [6] reported that HBP-1 mRNA in B-cells derived from BLV-infected cattle did not differ from those of BLV-negative animals. In a different study, Rosewick and colleagues used an ovine model of BLV infection to describe the expression of BLV-encoded miRNAs *in vivo* [5]. In that study, the authors used tumoral cell lines and ovine primary tumoral B-cells cultured *ex-vivo* to demonstrate an association between blv-miR-b4-3p expression and downregulation of HBP-1 but no association was found with PXDN. Apart from being the first description of BLV-encoded miRNAs, the different results obtained in previous studies suggest that the different experimental models might influence the outcomes, underscoring the necessity of studying those complex interactions in BLV natural host.

Cattle are the natural host of BLV and up to 10% of cattle infected with BLV develop B-cell lymphoma after several years of infection. During chronic infection with the BLV, proviral clones with low-level expression are selected by the strong immune pressure and it has been suggested that BLV-miRNAs might be involved in viral latency and cell malignant transformation [25]. In our study, we detected the expression of BLV-encoded blv-miR-b4-3p in three cattle that have been infected with BLV for more than one year, based on the detection of BLV-specific antibodies. Interestingly, we also detected the expression of BLV-pol mRNA in these animals, suggesting that BLV was not completely silent. Similarly, Kosovsky et al [9] reported expression of BLV-encoded miRNAs with variable levels of BLV-pol mRNA expression. Previously, our group reported spontaneous BLV transcriptional activation in cattle chronically infected with this virus [26]. Whether the three BLV (+) animals from the current study were in a transient phase of BLV-reactivation or BLV-transcripts were constantly expressed, couldn´t be determined at this time. Further studies are needed to determine correlation of BLV-encoded miRNAs and target mRNAs in the presence or absence of viral gene expression.

Our work has several limitations. In the first place, the reduced number of animals per group impacted the statistical potency of performed tests. We designed this work as a pilot study to test our hypothesis with the remarks of using the BLV naturally infected host. The high prevalence of BLV in Argentina´s dairy herds (>90%) posed an obstacle to finding true-negative BLV cattle for the present study. For that reason, we performed an exhaustive screening of candidate animals, to secure that the adult cattle in production that we used in the present study were negative for BLV. Future work with an increased number of animals per group will allow us to confirm and expand our observations. Although we have no evidence of a causal effect linking blv-miR-b4-3p with downregulation of PXDN, the association between the expression of blv-miR-b4-3p and the reduction of PXDN in BLV-infected animals was statistically significant. The higher expression of blv-miR-b4-3p compared to basal levels of miR-29a in these animals suggests a biological role of this particular BLV-encoded miRNA. Of note, the analogous host-miRNA miR-29a remained unchanged in BLV-infected animals, suggesting that the reduction of PXDN was at least partially directed by blv-miR-b4-3p. Finally,

previous *in silico*, *in vitro* and *ex vivo* analyses demonstrated a biological mechanism of blv-miR-b4-3p targeting different anti-oncogenic mRNA such as PXDN, supporting our observations [4].

In conclusion, we showed that blv-miR-b4-3p was highly expressed in cattle naturally infected with BLV and observed that PXDN was significantly sub-expressed compared to uninfected cattle. Considering the expression levels of bovine miR-29a were unchanged between BLV-infected vs uninfected cows, blv-miR-b4-3p could be responsible for the down-regulation of PXDN. The results presented here suggest that blv-miR-b4-3p might be involved in BLV tumorigenesis during natural infection with BLV in cattle.

## Supporting information

**S1 Data.**
(PDF)

## Acknowledgments

We acknowledge the Instituto de Virología e Innovaciones Tecnológicas (IVIT), Instituto Nacional de Tecnología Agropecuaria (INTA) and Consejo Nacional de Investigaciones Científicas y Técnicas (CONICET) for supporting our investigations.

## Author Contributions

**Conceptualization:** Hugo A. Carignano, Claudia Mongini, Diego D. Gonzalez, Juan P. Jaworski.

**Data curation:** Marcos I. Petersen, Juan P. Jaworski.

**Formal analysis:** Marcos I. Petersen, Juan P. Jaworski.

**Funding acquisition:** Juan P. Jaworski.

**Investigation:** Hugo A. Carignano, Juan P. Jaworski.

**Methodology:** Marcos I. Petersen, Juan P. Jaworski.

**Project administration:** Juan P. Jaworski.

**Supervision:** Hugo A. Carignano, Claudia Mongini, Juan P. Jaworski.

**Validation:** Marcos I. Petersen.

**Writing – original draft:** Marcos I. Petersen, Juan P. Jaworski.

**Writing – review & editing:** Hugo A. Carignano, Claudia Mongini, Diego D. Gonzalez, Juan P. Jaworski.

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
