## [Decision Letter · Decision Letter 0]

27 Dec 2022

PONE-D-22-32408Bovine leukemia virus encoded blv-miR-b4-3p microRNA is associated with reduced expression of anti-oncogenic gene in vivoPLOS ONE

Dear Dr. Jaworski,

Thank you for submitting your manuscript to PLOS ONE. After careful consideration, we feel that it has merit but does not fully meet PLOS ONE’s publication criteria as it currently stands. Therefore, we invite you to submit a revised version of the manuscript that addresses the points raised during the review process.

<ul> The authors provided novel and intriguing results by investigating the role of a potential oncogenic mechanism driven by Bovine leukemia virus infection. However, I would suggest a more in-depth discussion of the data. As known, miRNAs work in concert with each other to lead the pathological processes and their multiple targeted-mRNAs either. Since the authors pointed to the relevance of a unique genetic biomarker, I would encourage them to add some statements to highlight the importance of their work. Furthermore, I recommend citing more literature in the introduction. <li> A thorough proofreading of the English language is needed.

We look forward to receiving your revised manuscript.

Kind regards,

Elisabetta Pilotti

Academic Editor

PLOS ONE

Journal Requirements:

Reviewers' comments:

Reviewer's Responses to Questions

**Comments to the Author**

1. Is the manuscript technically sound, and do the data support the conclusions?

Reviewer #1: Yes

Reviewer #2: Partly

2. Has the statistical analysis been performed appropriately and rigorously? 

Reviewer #1: Yes

Reviewer #2: N/A

3. Have the authors made all data underlying the findings in their manuscript fully available?

Reviewer #1: Yes

Reviewer #2: Yes

4. Is the manuscript presented in an intelligible fashion and written in standard English?

Reviewer #1: Yes

Reviewer #2: Yes

5. Review Comments to the Author

Reviewer #1: 1. You must write P values for all figures (Figure 1, A (BLV-pol), B (viral blv-miR-b4-3p), C (host miR-29a), and E (HBP-1 mRNA)), since the graphical difference is not always so obvious, not only for Figure 1, D (PXDN mRNA).

2. Why didn't you measure DNA purity by the ratio of absorbance at 260/280 nm? This is one of the important indicators of DNA purity.

3. In the fourth sentence of the abstract: “… host ´s …” ─ should be “… host …”.

4. In the last sentence of the abstract: “… BLV-encoding miRNAs…” ─ don't generalize to everything BLV-encoding miRNAs, but only blv-miR-b4-3p: It will be more correct.

Line 41: “… RNA virus” ─ does not correspond to the adhered classification, instead you should write “Deltaretrovirus genus”. If you want to emphasize that this is the RNA virus, then it is better to write additionally: “According to the Baltimore classification system, BLV is the +ssRNA virus belonging to Group IV” [Baltimore D. Expression of animal virus genomes. Bacteriol Rev. 1971;35(3):235–41]

Lines 45: “… novo …” ─ should be “… de novo …”

Lines 45-46: Redundant information: “The novo synthesis of genomic viral RNA” and “viral transcripts” ─ rephrase this sentence.

Lines 48: “… persist …” ─ should be “… persists …”.

Lines 57: “... host ´s …” ─ should be “… host …”.

Lines 57-61: Where is the reference to these two sentences?

Lines 64: “… mRNA …” ─ should be “… mRNAs, …”.

Line 248: “… that BLV-encoding miRNAs might be involved in BLV tumorigenesis during natural infection” ─ I think that herein it is more correct not to focus on all BLV-encoding miRNAs, but only on blv-miR-b4-3p.

Reviewer #2: Dear Juan Pablo Jaworski,

The research paper entitled "Bovine leukemia virus encoded blv-miR-b4-3p microRNA is associated with reduced

expression of anti-oncogenic gene in vivo" provides a clue about the role of BLV miRNA in the development of cancer in infected animals. It is an important study and definitely adds up in the existing scientific knowledge about BLV. You are requested to improve the picture quality to 600dpi as the figures are blur and are not of good quality. Secondly discussion is week and requires further strengthening moreover the conclusion is lacking. very clear conclusion supporting your research work would be more impactful for the readers.

Regards

6. PLOS authors have the option to publish the peer review history of their article (what does this mean?). If published, this will include your full peer review and any attached files.

Reviewer #1: No

Reviewer #2: **Yes: **Nazish Bostan

---

## [Author Response · Author response to Decision Letter 0]

4 Jan 2023

Castelar, January 2nd 2023

PLOS ONE – Editorial Office

Academic Editor

Dear Elisabetta Pilotti

Thank you for the reviews and the opportunity to re-submit this manuscript entitled “Bovine leukemia virus encoded blv-miR-b4-3p microRNA is associated with reduced expression of anti-oncogenic gene in vivo” (PONE-D-22-32408). The manuscript has been modified in response to the Editor´s request and Reviewers´ helpful comments. On the following pages, we have addressed all of the comments from the reviewers. Reviewers´ comments are in bold and authors´ answers are in plain text below, prefaced by “AU” 

With these changes, my co-authors and I hope that you will find that the revised manuscript is now acceptable for publication in PLOS ONE.

Thank you for considering our manuscript for review.

Sincerely,

Juan Pablo Jaworski, DVM MSc PhD

Staff Scientist, Consejo Nacional de Investigaciones Científicas y Técnicas (CONICET)

Instituto de Virología, C.I.C.V. y A., Instituto Nacional de Tecnología Agropecuaria, Nicolás Repetto y De los Reseros s/n, Hurlingham (CP1686), Buenos Aires, Argentina.

Tel. /Fax: 54 11 4621 1447 (3400)

E-mail address: jaworski.juan@inta.gob.ar

Editorial Comments

The authors provided novel and intriguing results by investigating the role of a potential oncogenic mechanism driven by Bovine leukemia virus infection. However, I would suggest a more in-depth discussion of the data. As known, miRNAs work in concert with each other to lead the pathological processes and their multiple targeted-mRNAs either. Since the authors pointed to the relevance of a unique genetic biomarker, I would encourage them to add some statements to highlight the importance of their work. Furthermore, I recommend citing more literature in the introduction.

AU: The discussion has been strengthened highlighting the importance of HBP-1 and PXDN in cancer pathways. Additionally, as suggested by Reviewer 2 a conclusion was added at the end of the discussion highlighting the main findings of current work.

A thorough proofreading of the English language is needed.

AU: The text has been revised accordingly

AU: Checked 

AU: Modification of Data Availability Statement: In compliance with PLOS' data policy, all data required to replicate all study findings reported in this article (i.e., values behind the means, standard deviations, values used to build graphs, etc.) has been shared in Supporting Information files. This file (RAW data.PDF) is accessible by general software and has been included in the revised version of the manuscript.

AU: reference list has been revised and new references have been included as requested by Editor and Reviewer 2 in order to expand the introduction and discussion of the revised manuscript

1. Baltimore D. Expression of animal virus genomes. Bacteriol Rev. 1971 Sep;35(3):235–41. Available from: https://journals.asm.org/doi/10.1128/br.35.3.235-241.1971

10. Kincaid RP, Sullivan CS. Virus-Encoded microRNAs: An Overview and a Look to the Future. PLOS Pathog. 2012 Dec;8(12):e1003018. Available from: https://journals.plos.org/plospathogens/article?id=10.1371/journal.ppat.1003018

Reviewers' comments:

Reviewer #1:

1. You must write P values for all figures (Figure 1, A (BLV-pol), B (viral blv-miR-b4-3p), C (host miR-29a), and E (HBP-1 mRNA)), since the graphical difference is not always so obvious, not only for Figure 1, D (PXDN mRNA).

AU: P values were added for figure 1.C; 1.D and 1.E (miR-29a, PXDN and HBP-1, respectively). No P values were calculated for blv-miR-b4-3p and BLV-pol since none of these targets were detected in BLV-negative animals (BLV-).

2. Why didn't you measure DNA purity by the ratio of absorbance at 260/280 nm? This is one of the important indicators of DNA purity.

AU: Thank you for pointing out this mistake/omission. As now stated in line 110 “DNA purity was assessed by the ratio of absorbance at 260/230 nm and 260/280 nm” for the detection of salt and protein contaminants, respectively. 

3. In the fourth sentence of the abstract: “… host ´s …” ─ should be “… host …”.

AU: corrected

4. In the last sentence of the abstract: “… BLV-encoding miRNAs…” ─ don't generalize to everything BLV-encoding miRNAs, but only blv-miR-b4-3p: It will be more correct.

AU: we agree with this observation and it is now corrected as suggested

Line 41: “… RNA virus” ─ does not correspond to the adhered classification, instead you should write “Deltaretrovirus genus”. If you want to emphasize that this is the RNA virus, then it is better to write additionally: “According to the Baltimore classification system, BLV is the +ssRNA virus belonging to Group IV” [Baltimore D. Expression of animal virus genomes. Bacteriol Rev. 1971;35(3):235–41]

AU: Now corrected as follows “Bovine leukaemia virus (BLV) belongs to the Deltaretrovirus genus, Orthoretrovirinae subfamily and Retroviridae family. BLV is related to the human T-cell lymphotropic virus (HTLV-1 and -2). According to the Baltimore classification system BLV is a +ssRNA virus (Group IV [Baltimore D. Expression of animal virus genomes. Bacteriol Rev. 1971;35(3):235–41])”. New citation added to reference list.

Lines 45: “… novo …” ─ should be “… de novo …”

Lines 45-46: Redundant information: “The novo synthesis of genomic viral RNA” and “viral transcripts” ─ rephrase this sentence.

AU: the sentence has been rephrased as follows: “The synthesis of viral transcripts and proteins is a hallmark of this stage.”

Lines 48: “… persist …” ─ should be “… persists …”.

AU: fixed

Lines 57: “... host ´s …” ─ should be “… host …”.

AU: corrected 

Lines 57-61: Where is the reference to these two sentences?

AU: A reference was added to support both statements “Virus-Encoded microRNAs: An Overview and a Look to the Future” Kincaid and Sullivan, Plos Pathogens (REVIEW) Dec 2012; Vol: 8, Issue: 12.

Lines 64: “… mRNA …” ─ should be “… mRNAs, …”.

AU: fixed 

Line 248: “… that BLV-encoding miRNAs might be involved in BLV tumorigenesis during natural infection” ─ I think that herein it is more correct not to focus on all BLV-encoding miRNAs, but only on blv-miR-b4-3p.

AU: The statement has been modified as follows: “Moreover, since PXDN has been linked to anti-tumoral pathways in B-cells, the downregulation of PXDN expression by blv-miR-b4-3p might be involved in BLV tumorigenesis in cattle during natural infection with this virus.”

Reviewer #2: 

Dear Juan Pablo Jaworski,

The research paper entitled "Bovine leukemia virus encoded blv-miR-b4-3p microRNA is associated with reduced expression of anti-oncogenic gene in vivo" provides a clue about the role of BLV miRNA in the development of cancer in infected animals. It is an important study and definitely adds up in the existing scientific knowledge about BLV. 

You are requested to improve the picture quality to 600dpi as the figures are blur and are not of good quality. 

AU: FIGURE 1 quality has been improved as requested

Secondly discussion is week and requires further strengthening moreover the conclusion is lacking. very clear conclusion supporting your research work would be more impactful for the readers.

Regards

AU: The discussion has been strengthened highlighting the importance of HBP-1 and PXDN in cancer pathways. Additionally, as suggested by this Reviewer a conclusion was added at the end of the discussion highlighting the main findings of the current work

---

## [Editor Report · Decision Letter 1]

20 Jan 2023

Bovine leukemia virus encoded blv-miR-b4-3p microRNA is associated with reduced expression of anti-oncogenic gene in vivo

PONE-D-22-32408R1

Dear Dr. Jaworski,

We’re pleased to inform you that your manuscript has been judged scientifically suitable for publication and will be formally accepted for publication once it meets all outstanding technical requirements.

Kind regards,

Elisabetta Pilotti

Academic Editor

PLOS ONE
---

## [Editor Report · Acceptance letter]

24 Jan 2023

PONE-D-22-32408R1 

Bovine leukemia virus encoded blv-miR-b4-3p microRNA is associated with reduced expression of anti-oncogenic gene *in vivo*

Dear Dr. Jaworski:

I'm pleased to inform you that your manuscript has been deemed suitable for publication in PLOS ONE. Congratulations! Your manuscript is now with our production department. 

Kind regards, 

on behalf of

Dr. Elisabetta Pilotti 

Academic Editor

PLOS ONE